# A systematic review and evidence synthesis of non-medical triage, self-referral and direct access services for patients with musculoskeletal pain

Opeyemi O. Babatunde[1]*, Annette Bishop[1], Elizabeth Cottrell[1], Joanne L. Jordan[1], Nadia Corp[1‡], Katrina Humphries[1‡], Tina Hadley-Barrows[1‡], Alyson L. Huntley[2‡], Danielle A. van der Windt[1]

1 Arthritis Research UK Primary Care Centre, School of Primary, Community & Social Care, Keele University, Keele, United Kingdom, 2 Centre for Academic Primary Care, Bristol Medical School, University of Bristol, Bristol, United Kingdom

☯ These authors contributed equally to this work.
‡ These authors also contributed equally to this work.
* o.babatunde@keele.ac.uk

**Data Availability Statement:** All relevant data are within the paper and its Supporting Information files.

## Abstract

### Introduction

The demand for musculoskeletal (MSK) care is rising, and is a growing challenge for general practice. Direct access to physiotherapy and other healthcare services may offer appropriate care for MSK pain patients but there is uncertainty regarding the effectiveness or efficiency of this approach in practice. This study aimed to review the evidence regarding characteristics, outcomes, barriers and facilitators of MSK triage and direct access services.

### Methods

A comprehensive search of eight databases (including MEDLINE, EMBASE, and Cochrane library) up to February 2018 was conducted to identify studies (trials, cohorts and qualitative evidence) on direct access services for MSK in primary care settings. Using predefined inclusion and exclusion criteria, titles, abstracts, and subsequent full texts were independently screened by reviewers. Methodological quality of eligible studies was assessed using the mixed methods appraisal tool, and extracted data regarding study characteristics and results were independently reviewed. A narrative synthesis and grading of evidence was undertaken. Approaches to MSK triage and direct access were profiled along with their respective outcomes of care relating to patient-oriented and socioeconomic outcomes. Barriers and facilitators of each model of direct access services were also highlighted.

### Results

9010 unique citations were screened, of which 26 studies were eligible. Three approaches (open access, combination and service pathway models) to MSK triage and direct access shared similar goals but were heterogeneous in application. MSK patients using direct

**Funding:** This work was funded by the National Institute for Health Research (NIHR) School for Primary Care Research (Evidence Synthesis Working Group project reference 390). OB is a fellow of the NIHR academy and is supported through an SPCR seedcorn funding. EC is an NIHR Clinical Lecturer in Primary Care. The views expressed are those of the author(s) and not necessarily those of the NIHR or the Department of Health and Social Care. The funders had no role in study design, data collection and analysis, decision to publish, or preparation of the manuscript.

**Competing interests:** The authors have declared that no competing interests exist.

access showed largely similar characteristics (age, sex and duration of symptoms) compared to GP-led care, although they were often younger, slightly more educated and with better socio-economic status than patients seen through GP-led care. Although many studies showed limitations in design or methods, outcomes of care (patient oriented outcomes of pain, and disability) did not show large differences between direct access and GP-led care. In most studies direct access patients were reported to have lower healthcare utilisation (fewer physiotherapy or GP consultations, analgesics or muscle relaxants prescriptions, or imaging procedures) and less time off work compared to GP-led care.

## Discussion

This study provides insight into the current state of evidence regarding MSK triage and direct access services and highlights potential implications for future research, healthcare services planning, resource utilisation and organising care for MSK patients in primary care. There is consistent, although limited, evidence to suggest that MSK triage and direct access services lead to comparable clinical outcomes with lower healthcare consumption, and can help to manage GP workload. However, due to the paucity of strong empirical data from methodologically robust studies, a scale up and widespread roll out of direct access services cannot as yet be assumed to result in long term health and socio-economic gains. PROSPERO-ID: CRD42018085978.

## Introduction

Musculoskeletal (MSK) pain problems including low back pain (LBP), shoulder pain, neck pain, knee pain and widespread pain are leading causes of years lived with disability globally [1]. Mostly managed in primary care, they are the second most common reason for sickness certification, resulting in an estimated 10 million lost working days and up to 50 million consultations per year in the United Kingdom (UK) [2, 3]. Partly due to ageing populations and an increasing prevalence of obesity, the demand for musculoskeletal care is set to rise, and is a growing challenge for primary care globally [1]. In the UK for instance, these population changes are compounded by a reducing general practitioner (GP) workforce and increasing patient demand. Evidence shows that MSK problems are long-term conditions, often following a course characterised by relapses and recurrences [4], and that many patients with MSK conditions presenting to GPs will eventually be referred onwards to physiotherapists and other non-medical professionals [5, 6]. As such, patient direct access to physiotherapy, musculoskeletal triage and first contact management by suitable non-medical professionals may offer appropriate, effective and efficient solutions to both getting patients seen at the right time by the most appropriate healthcare professional; and proactively managing rising demand over time, reducing the burden of MSK management on existing GP services.

Patient direct access (also known as self-referral) for MSK care is a system of access in which patients are able to refer themselves directly to a non-GP first contact professional without having to see anyone else first, or without being told to refer themselves by a medical practitioner. In over half of EU member states and most parts of the US, patients can self-refer to physiotherapists but there are variations as to how direct access services are being operationalised in these countries. It is also not clear which of these service models is most clinically and economically effective. Currently, in the UK, there is a policy drive to broaden the professional

workforce delivering primary care [7–9]. This has resulted in multiple service models being delivered within primary care as an alternative to the traditional GP-led model. These include, first contact practitioners, who are physiotherapists with extended skill sets and who assess and provide the management plan for patients with MSK conditions, through to in-practice nurse practitioners, physiotherapists, and physician associates who may provide a first-contact service for patients presenting to their primary care practice. A systematic review which investigated substitution of doctor roles by physiotherapists, suggested patient clinical outcomes are similar and satisfaction is the same or better compared to consulting a physician, but the findings were based on research primarily from specialist orthopaedic services [10]. Several uncertainties about, and barriers to adoption of non-GP first contact healthcare professionals have been identified related to, for example, volume and characteristics of patients using such services (with some studies showing self-referral services were only used by specific subgroups of patients); or the perception that only physicians can independently diagnose and treat patients presenting with a new MSK condition. However, there is currently no robust evidence synthesis, systematically summarising current knowledge on the various direct access/self-referral service models, and associated barriers and facilitators for the management of MSK conditions in primary care settings.

Therefore, in order to inform future practice, legislation and/or organisation of healthcare, specific objectives of this study were to:

a. determine the characteristics of patients making use of MSK triage and/or non-medical direct access services;

b. describe currently available models of MSK triage and direct access to non-medical first contact services in primary care settings as well as the barriers and facilitators associated with such models;

c. synthesize evidence regarding outcomes of MSK triage or non-medical direct access services in relation to patient outcomes (pain, disability, work absence and sickness certification), safety (e.g. missed red-flag diagnoses), socio-economic and health care costs (consultations, prescriptions, tests, referrals, and impact on GP workload/services).

Addressing the stated aims of this review will help to understand currently available MSK triage and direct access services, ascertain its' effectiveness, and explore ways by which services (if effective) could be improved and extended to all, thereby decreasing health inequality among patients with MSK pain conditions.

## Methods

### Patient and public involvement

A patient and public involvement and engagement (PPIE) Research User Group (RUG; n = 8) advised the review team during the conduct of this review. When consulted on the objectives and design of this study, the RUG members, who are patients with present or previous experiences of MSK conditions, validated the appropriateness of the research question and study design. Specifically, RUG members emphasised the need to extract pertinent information from included papers regarding the accessibility of MSK triage/self-referral and the impact of such services on GP workload/services.

### Systematic review protocol and registration

A protocol, outlining the review questions, and planned synthesis was developed a priori and registered with the international prospective register of systematic reviews (PROSPERO-ID:

CRD42018085978). A lay summary of the review was developed and is available on the website of the Evidence Synthesis Working Group [https://www.spcr.nihr.ac.uk/eswg/urgent-care-interface]. This review was conducted and reported in accordance with the Preferred Reporting Items for Systematic Reviews and Meta-Analyses (PRISMA) statement [11].

## Information sources and search strategy

An information specialist (NC) developed the search strategy with input from the study team involving clinicians and academics with MSK expertise (please refer to supplementary file, S1 Table for the full Medline search strategy). A comprehensive search of 8 databases (MEDLINE, EMBASE, AMED, CINAHL, PsycINFO, Cochrane library, Web of Science and Pedro–from their inception to February 2018) was conducted to identify studies (trials, cohorts and qualitative studies) evaluating triage and/or non-medical direct access services in primary/community care settings for patients with MSK conditions. This was complemented by hand searching of references of eligible full texts. A regular current awareness search for newly published studies was set up and was used to alert authors to new publications in the area.

## Eligibility and study selection

To be eligible for inclusion, studies had to evaluate primary care, musculoskeletal triage and/or non-medical direct access services for adults (18 years and over) with MSK conditions in terms of clinical outcomes (e.g. pain, functional disability), socio-economic outcomes (costs of care, healthcare utilisation), and/or facilitators and barriers. Such services had to be set in primary/community care, but not led, or referred to, by GPs. In this way, services considered within this review were a direct alternative to traditional GP-led care. Any non-GP (healthcare professional) delivering the service was eligible. Studies were included if they were experimental (e.g. randomised trials, comparative cohort studies, before-after designs) or non-experimental (prospective or retrospective observational cohort studies, qualitative studies, cross-sectional surveys) in design. There was no restriction to the length of follow-up, language and publication date (please see supplementary file, S2 Table for detailed eligibility criteria).

Title screening based on the eligibility criteria was piloted for a random selection of studies (n = 200) by pairs of reviewers. Conflicts (n = 32) were then discussed and resolved in a meeting involving the whole team in order to establish consistency of interpretation and application of rules regarding the eligibility criteria. Subsequent title screening was performed by reviewers, excluding studies that clearly did not meet the eligibility criteria. For both abstracts and full text selection stages, reviewers independently evaluated the eligibility of each of the identified studies in pairs. Disagreements were resolved through discussion or by third reviewer adjudication.

## Data items and data collection process

A customised data extraction tool was developed and used to extract details, for each included study regarding: study design (experimental and non-experimental procedures as applicable); study setting; recruitment/sampling; aims of the study; inclusion criteria; baseline characteristics of the study sample (age, gender, diagnosis, and pain duration); details of interventions (type of service, healthcare professionals involved, triage only or triage with diagnosis and treatment); and outcome assessments: patient specific (e.g., pain, function)/ generic (e.g., return to work, QOL); safety (e.g., missed red-flag diagnoses); health care-costs e.g., direct and indirect costs of MSK triage and direct access service; socioeconomic e.g., demand, impact on patients and GP services.

Expressed and/or perceived barriers and facilitators of MSK triage and direct access by patients and various health professionals within included studies were extracted. Where available, data relating to the fidelity of the MSK triage and direct access service described in each study were also captured. Specifically, this relates to the extent to which MSK triage and direct access services were delivered as planned; and if any strategies (e.g. longer/shorter duration of consultations, training of service providers, protocols/algorithms) were used to maintain or improve adherence, uptake, and adequacy of the support systems for these services.

The consistency of data extraction was piloted prior to the main extraction on three papers (picked at random considering different study designs included in the review). Subsequently, data extraction for each included study was performed and checked for completion and accuracy by pairs of reviewers (OB, AB, EC, NC, AH, KH, THB, DvdW). Discrepancies in extracted data were resolved by the independent adjudication of a third reviewer.

## Study quality assessments

The methodological quality of included studies was assessed using the Mixed Method Appraisal Tool (MMAT) [12]. The MMAT criteria were designed to concurrently appraise qualitative, quantitative, and mixed method studies for large and complex systematic reviews and is well suited for the assessment of complex interventions that are context-dependent and process-oriented, such as triage and direct access for healthcare services. Items were scored as yes, no or unclear (depending on if criteria were fully met, not met or there was insufficient information in the report to judge, respectively) at the individual study level and overall (across studies). Discrepancies were resolved through discussion between pairs of reviewers or by a third reviewer.

## Data synthesis and analysis

A random effects meta-analysis was planned but was not conducted due to lack of suitable, homogeneous outcome data across studies evaluating similar services.

A narrative synthesis involving a three-stage analysis was conducted linked to the three objectives of the review. The first stage (objective a) involved characterising the patients using the service(s) detailed within each study. The second stage analysis (objective b) first focussed on the development of the classification of MSK triage and direct access models. Specifically, studies were sorted and grouped based on the reported characteristics of services and their approach to triage and/or direct access service. An initial sorting phase was undertaken by three reviewers (OB, AB, EC) with subject knowledge of MSK care in primary/community care settings and systematic review methods expertise, who suggested groupings based on approaches used for triage, direct access, or self-referral. The grouping of the services was further discussed, modified and ratified by the review team (OB, AB, EC, DvdW, KH, THB, AH), which resulted in a classification of services based on available evidence from the included studies.

Next, where available, expressed and perceived barriers and facilitators of each service as described within each of the included studies were profiled and aggregated, reflecting patient and health care professional perspectives and/or experiences, as well as organisational issues. Evidence regarding perceived barriers and facilitators of each of the classified MSK triage and direct access service models were subsequently mapped and incorporated into the evidence for each service type/models, as supported by data from the studies.

The third stage (objective c) described and synthesised the outcomes of MSK triage and/or direct access services in relation to patient outcomes. Evidence of the effectiveness of MSK triage and direct access services for each clinical and socioeconomic outcome was synthesised

and graded using a modified GRADE (http://www.gradeworkinggroup.org/) approach, taking into account the hierarchy of evidence, quality of the evidence, level of precision, and consistency of results across the studies (please see S3 Table for details) [13].

Subsequently, evidence regarding outcomes of MSK triage or direct access services in relation to patient outcomes (pain, disability, work absence and sickness certification), safety (e.g. missed red-flag diagnoses), socio-economic and health care costs (consultations, prescriptions, tests, referrals, and impact on GP workload/services) were graded using the criteria as described above and a narrative synthesis was subsequently presented, indicating the strength of the evidence as very weak, limited, moderate, or strong.

## Results

### Study flow and characteristics of included studies

The literature search yielded 9010 unique citations, of which 405 articles were selected for full text review. No new studies were identified by hand searching of the references of included full texts or grey literature. Forty-five full text articles met the eligibility criteria and were subjected to quality assessment and data extraction. Two most common reasons for exclusion of full text articles were that the triage and/or direct access service was not primarily offered for MSK conditions (or results were not separately described for patients with MSK conditions); or where telemedicine was used as a substitute, or to augment usual GP care for MSK conditions, but did not involve triage or direct access services. Nineteen articles were further excluded from the review as they were later judged to be duplicates or additional reports of included studies (n = 14) or they presented perceptions of patients or stakeholders regarding "hypothetical" situations where patients have not been in actual receipt of care via direct access (n = 5). Twenty-six studies evaluating direct access services for MSK patients were subsequently synthesised in this review. The detailed study flow chart and summary of reasons for exclusion are presented in Fig 1.

Characteristics of the 26 studies are presented in Table 1. With the exception of four trials [14–17] and one qualitative study [18], which explored patients' experiences of direct access through interviews; included studies were mostly observational by design (8 before and after service evaluations [19–27], including 5 cohorts [28–32]; 4 surveys [33–36]; and 4 cross-sectional studies [37–40]). About half of the studies (n = 12) were conducted in America [14, 15, 20, 21, 26–28, 30, 33, 34, 37, 40], and 10 in the United Kingdom [16–19, 22–24, 29, 31, 35, 36]. Others (n = 3) were conducted in Europe–specifically in Netherlands [25] and Sweden [32, 40]. The only study to be conducted in a low income country (Afghanistan) was related to an American armed forces medical centre and reported on a service which was solely for servicemen and associated personnel [39]. Studies recruited participants mostly from the community or primary care settings, and all but one study (an MSK triage service to trained nurse professionals) [14], studied direct access to physiotherapist-led services for MSK conditions compared to GP-led care.

### Study quality

For many aspects of the quality criteria assessment, as much as half of the responses were either a "no" or "can't tell" where studies clearly did not meet the expected criteria or due to lack of clarity in the report to facilitate clear judgement of study quality. Among the four trials, only two were judged to have carried out adequate randomisation process, gained comparable samples at baseline and also controlled the application of intervention protocols [15, 16]. The trial by Greenfield et al. was assessed to have sufficiently met methodological quality criteria on only one domain, having presented complete outcome data [14]. One trial was quasi

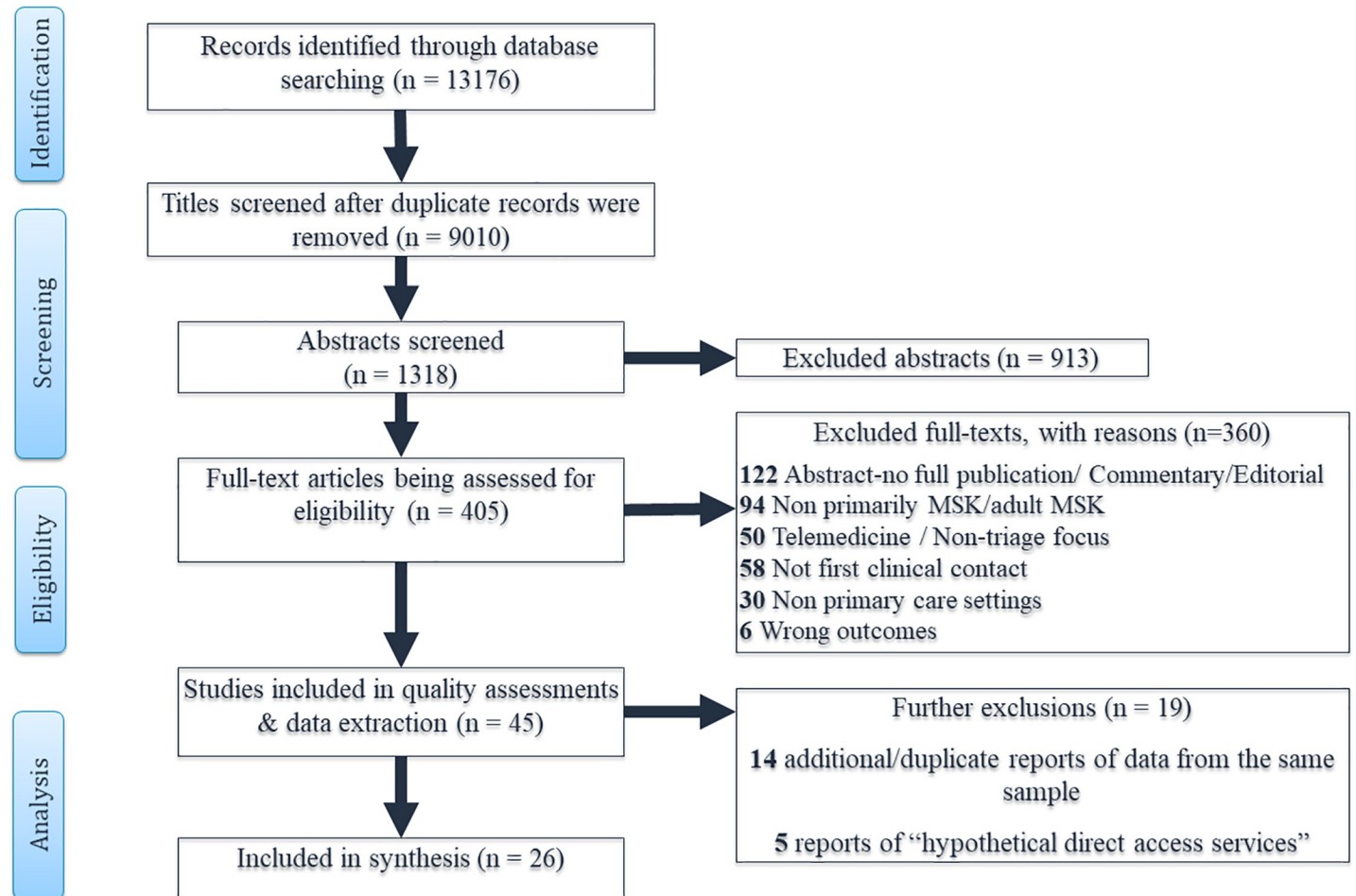

**Fig 1. Study flow chart.**

experimental in design and was therefore assessed as a non-randomised quantitative study [17]. Of the remaining non-trial designs (n = 22), over 70% (n = 16) were assessed as having recruited appropriate participants sufficiently representative or relevant to the primary research questions. The rest (n = 6) generated a "no" response to this assessment criterion or did not include sufficient details in the report to facilitate a clear judgement in this regard. Noticeably, confounders and other factors associated with outcome were not always accounted for in the study design and analysis (n = 15), and studies mostly failed to report complete outcome data for all participants (n = 13). Results of study quality appraisal using the MMAT tool are shown in S4 Table.

## Characteristics of patients attending MSK triage and direct access—study objective 1

Overall, this systematic review presents data involving a total of 62,775 patients who accessed care for their MSK conditions through direct access to non-medical professionals compared to 57,501 patients treated for MSK conditions though usual GP-led care. Not all studies involved direct comparisons, as some (n = 9) focussed solely on direct access patients [19–23, 26, 27, 29, 34, 39]. In addition, six studies [18, 31, 33, 35] explored the views, attitudes and experiences of

**Table 1. Characteristics of included studies.**

| First Author /Year of publication Country | Study Aim(s) | Study setting | Study Design | Eligibility criteria | Sample size (proportion of males) | mean age (SD) | Diagnosis (where specified) | Chronicity of Symptoms in weeks. Mean(SD) | Comments on fidelity / summary of study findings |
|---|---|---|---|---|---|---|---|---|---|
| Badke et al 2014 USA | To compare cost and utilization variables when patients were seen by physical therapists with and without a physician referral | University health centre | retrospective cohort. | Included: MSK episodes of care over 2 years Excluded: inpatient stays in the 2 yrs window; surgery including those requiring postoperative rehabilitation; having both PR and DA care; patients with Medicare, Medicaid, or workers compensation insurance. | DA: n = 252 46.4% males PR: n = 169 40.2% males | DA:41.9 (13.9) PR: 39.8 (16.6) | spinal impairments and sports injuries (e.g. backache, lumbago, joint pain and stiffness, neck pain, and shoulder dysfunction/pain). | DA: 20.4 (48.6) PR: 18.5 (28.9) | Data source: billing data. No difference in age, sex, diagnosis, chronicity of symptoms, treatment duration between DA & PR patients. Overall, mean physical therapy visits was significantly higher for PR (5.4 ±3.2) than DA (3.9 ±3) |
| Mintken 2015 USA | To determine occurrence of adverse events related to physiotherapy management of patients via direct access. | University health centre | retrospective analysis: Before & after study | All patient visits to the direct access clinic for MSK over a 10-year period. | 12976 (60% males). 98%, 2% of DA patients were university students and staff respectively. | NR | Ankle/foot 25%, knee 33%, hip/thigh 7%, hand/wrist 9%, elbow/forearm 3%, shoulder 16%, spine 4%, others 3% | NR | Data source; clinic personnel files, electronic health records, and risk management office. Concerns only DA patients, therefore, no comparison data. PTs average years of experience (8.8 ± 5.9) involved in DA. Most of the PTs obtained further certification and doctoral degrees during the time frame of the study. No adverse events or professional revocation of licence recorded. |
| Moore 2005 USA | To determine risk of adverse events related to physiotherapy management of patients via direct access in military Health settings | Occupational healthcare | retrospective analysis: Before & after study | All patient visits to the direct access clinic for MSK over a 40-month period | DA: n = 22, 910 Proportion of males NR. | NR | Mainly common musculoskeletal injuries (e.g., retro-patellar pain syndrome, ankle sprains, shoulder impingement, low back pain) + others (non-MSK) | NR | Nearly all (98%) of the PTs involved in DA have higher degrees (masters & doctoral) and obtained further certification and specialty training in Neuro-MSK evaluation. Limited data. Study finds MSK patients seen via direct access to PTs in military health care facilities are at minimal risk of serious adverse events. |

(*Continued*)

**Table 1.** (Continued)

| First Author /Year of publication Country | Study Aim(s) | Study setting | Study Design | Eligibility criteria | Sample size (proportion of males) | mean age (SD) | Diagnosis (where specified) | Chronicity of Symptoms in weeks. Mean(SD) | Comments on fidelity / summary of study findings |
|---|---|---|---|---|---|---|---|---|---|
| Ojha 2015 USA | to determine early outcomes of direct access PT for university employees | University health centre | retrospective analysis: Before & after study | University employees with acute injuries <3 months after onset Excluded patients with prior consultations/ referral for the same condition; previous surgery/psychiatric diagnosis; red flag symptoms. | DA: n = 10 | NR | a primary complaint that was potentially of neuro-musculoskeletal aetiology | All: <12 weeks | Limited data from small pilot study over 1-week period. Involves a single therapist. Concerns only DA patients. therefore, no comparison data. Direct access PT was associated with positive clinical outcomes and low total cost. |
| Denninger 2018 USA | To compare total claims and patient outcomes for MSK care between DA and PR services. | Community | retrospective cohort | Employees and adult dependents of a community health system, 18 years or older, with neck or back pain Excluded patient data for unplanned discharge or attended less than 6 sessions with no follow-up data. | DA: n = 171; 41% males PR: n = 276; 27% Males | DA: 47.5 (10.8) PR: 44.9 (12.3) | Neck or back complaints | DA: Acute:16%, subacute:20%, chronic: 63% PR: Acute:32%, subacute:14%, chronic: 54% | Data Source: Patient Outcomes Registry, & US Department of Health and Human Services Agency for Healthcare Research and Quality in the Registry of Patient Registries Healthcare utilisation costs may not be fully accounted for as 24 patients were missing from the study's flow chart Except for chronicity of symptoms, multiple pain sites, no significant difference in age, sex, diagnosis, treatment duration between DA & PR patients. |
| Swinkels 2014 Netherlands | to investigate the outcomes of a national service involving direct access to physical therapy for MSK patients over 5 yrs, compared to referral-based physical therapy. | Primary care/ outpatients | retrospective analysis: Before & after study | Codes in the electronic health record for MSK pain. | DA: n = 4,941 47% males PR: n = 7,077 42% males. | DA: 47.0 (16.3) PR: 50.3(17.9) | Back pain, neck, shoulder, and knee complaints | DA: <7 d:22% 1–12weeks: 58% 12-56weeks:8% >56weeks:11% PR: <7 d: 9% 1–12weeks:57% 12-56weeks:16% >56weeks:17% | Data source: electronic health records Study founds significant associations with engagement with direct access for males, middle/younger aged, higher education, previous physical therapy, recurrent back pain, acute episodes of pain <7days and less severe back pain. |

(*Continued*)

**Table 1.** (Continued)

| First Author /Year of publication Country | Study Aim(s) | Study setting | Study Design | Eligibility criteria | Sample size (proportion of males) | mean age (SD) | Diagnosis (where specified) | Chronicity of Symptoms in weeks. Mean(SD) | Comments on fidelity / summary of study findings |
|---|---|---|---|---|---|---|---|---|---|
| Pendergast et al 2012 USA | To compare patient profiles and healthcare use for self- and physician referred patients. | Hospital/ Rehabilitation | Cross-sectional analysis | 18-64yr beneficiaries of private insurance who accessed Physiotherapy care | DA: n = 17,497; 41.4% males PR: n = 45,210 40.85% males | DA: 43.5 (13.12) PR: 45.9 (12.62) | Arthritis, Spine pain, Sprain/strain, others | NR | Data source: Five years' private health insurance claims data Self-referred group was slightly younger, had fewer PT visits. |
| McGill et al 2013 Afghanistan | To compare efficiency and effectiveness of a physical therapist functioning as a MSK primary care provider compared to family practice physicians | Hospital/ Rehabilitation | Cross-sectional analysis | Active-duty or civilian contract personnel >18 years of age with MSK complaint. Excluded fractures, dislocations, or trauma where deformity is present, fevers or pain of a non-mechanical, non-musculoskeletal origin | All: n = 149, 84% males | All: Median age 29 (range 19-54yr) | Predominantly lumbar and knee pain but all main extremity joint sites were represented. | NR | Military setting Data Source: Medical records Lack of clear comparison data for participant demographics and outcomes. Data mostly relates to immediate aid & relief. Unclear if patients were followed up and if DA later seen by Physician. |
| Mitchell et al 1997 USA | To evaluate resource use and cost of direct access to physical therapy compared to physician referral | Unclear | Cross-sectional analysis | Working age adults who had at least one physical therapy claim during Jan 1990 to Dec 1993 Excluded persons eligible for Medicare (age 65 years and older) | DA: n = 252 PR: n = 353 | NR | Non-specified: Acute MSK diagnosis | NR | Data Source: Claims data No comparison data for DA/PR patient demographics. Possible errors associated with validity of claims and patient clinical characteristics. Excluded people with multiple comorbidities, chronic MSK conditions, and 65 years and over. **NB**: Private health care insurance system. |
| Bishop et al 2017 UK | To investigate the feasibility of a patient self-referral pathway to physiotherapy | Primary care | Cluster randomised trial | Patients aged 18yr or older presenting to their General Practice or physiotherapy service with a MSK condition Excluded patients undergoing palliative care, had severe learning disabilities, non-ambulatory | DA: n = 142, 48% Males PR: n = 553, 43.8% Males | DA: 56.5 (14.7) PR: 58.6 (14.6) | Non specified MSK | >6 weeks: | Pilot trial data only. Service based focus on organisation/provision of direct access. Increased uptake of DA in intervention practices. No difference in clinical and cost outcomes for DA and PR patients. |

(*Continued*)

**Table 1.** (Continued)

| First Author /Year of publication Country | Study Aim(s) | Study setting | Study Design | Eligibility criteria | Sample size (proportion of males) | mean age (SD) | Diagnosis (where specified) | Chronicity of Symptoms in weeks. Mean(SD) | Comments on fidelity / summary of study findings |
|---|---|---|---|---|---|---|---|---|---|
| Mallet 2014 UK | To access viability, cost effectiveness and patient benefit of DA to MSK services | Primary care | Prospective before & after service evaluation analysis. | MSK conditions | *DA = 105, PR = 89 | NR | Non-specified MSK, mostly spinal pain | DA: Mean 3.55, ±2.7 days PR: Mean 30.99, ±15.4, days | a higher uptake of DA by women, patients with more acute symptoms (<1 month). Many patients in self-referral pathway felt satisfied with care. |
| Bornhöft 2015 Sweden | to investigate effects of MSK triage on utilization of medical services. | Primary care | Cross-sectional analysis | Patients 16–64 years with MSK. Excluded non-MSK, recent prior visit to GP/ therapist for same problem. | DA: n = 656, 47.9% males PR: n = 1673, 40.2% males | DA: 34.4 (11.5) PR: 40.8 (12.3) | MSK including: back, spine, neck, upper/lower -extremity pain problems. | DA: acute<12wks: 50.5%, chronic>12wks:32.5%, Mixed:17.0% PR: acute:48.7%, chronic:38.9%, Mixed:12.4% | Data source: patient medical records Differences in demographics. Initial screening/ triage by nurses could have resulted in younger, healthier patients to DA. |
| Holdsworth 2007/2008 UK | Compare the demographic and clinical outcomes of self-referral to physiotherapy vs. usual GP care. Clinician and patient views of DA services | Primary care | Quasi experimental (trial) + evaluation | Adults, registered at a participating practice, who referred or self-referred to physiotherapy over 1-year study period. Excluded routine antenatal care & Hospital consultant referrals | DA: n = 1190, 38.6% males PR: n = 1795, 42% males 97 GPs, 64 PTs | DA: 53.0 (16.6), and 51.0 (15.5- Physician suggested), PR: 53.0 (16.7) | Low back, Neck, Lower limb, Shoulder, Knee, Upper limb, Multiple sites and others | DA: <6 wks: 51% 7-12wks: 17% >12wks: 32% PR: <6 wks:23% 7-12wks: 16% >12wks: 61% | Large trial involving 26 practices but issues with missing data. Direct access pats who self-referred were slightly different from physician suggested referrals. Study found no significant differences in gender or age for DA and PR patients. However, DA patients were more likely to have had less duration of symptoms up till the time of being seen by a Physiotherapist. |
| Phillips 2012 UK | To evaluate the feasibility and cost-effectiveness of the Occupational Health Physiotherapy Pilot Project | Occupational healthcare | prospective cohort | Employed by participating organisation, MSK condition | All (DA): n = 486, 36% males | 43.1(10.45) | All MSK disorders | 56.12 months (SD 91.1) | Pilot only, No comparison data Demand for telephone advice was very low. Follow-up at 3 months was 41% although the authors state responders did not differ from non-responders. Measured several other outcomes but did not report these. |

*(Continued)*

**Table 1.** (Continued)

| First Author /Year of publication Country | Study Aim(s) | Study setting | Study Design | Eligibility criteria | Sample size (proportion of males) | mean age (SD) | Diagnosis (where specified) | Chronicity of Symptoms in weeks. Mean(SD) | Comments on fidelity / summary of study findings |
|---|---|---|---|---|---|---|---|---|---|
| Greenfield 1975 USA | To ascertain effectiveness of a nurse-administered protocol for low back pain, | Community | ≠Trial | Adult patients who presented to the clinic with the complaint of low back pain. Protocol was not applicable to patients who had traumatic injury, auto accident or fall | DA (Nurse): n = 222, 48% males PR: n = 197, 44.7% males | NR | low back pain | NR | Non-randomised, highly selective sample for Nurse-led management. Age and gender fairly balanced across both groups at baseline. A relatively high proportion of Nurse led protocol patients were subsequently referred for physician management. |
| Goodwin 2016/Moffat 2017 UK | To evaluate the clinical effectiveness, patient satisfaction and economic efficacy of a physiotherapy service providing musculoskeletal care as an alternative to GP care. To understand what staff thought of self-referral | Primary care | prospective analysis of patient cohort/ qualitative service evaluation with staff. | Patients presenting at participating general practices with MSK Volunteered staff (n = 13) | All (DA): n = 123 | NR | Non-specified MSK conditions | Practice 1: <4 wks. 36%, > 64%. Practice 2: < 4 wks. 38%, >4 weeks 63%. | Hypothetical comparison to GP led-care retrospectively. Study data concerns DA patients and relates to single consultations. So significant difference in patient demographics. Not all costs included and impact of case mix not considered. Feasibility was based only on rate of uptake. No qualitative methods were used to establish reasons for non-uptake particularly for the low levels of telephone advice. |
| Overman et al 1988 USA | To compare outcomes of physical therapy first contact with physician first contact. | Community | ≠Trial | LBP Excluded non-LBP; non-consenting | DA: n = 107 PR: n = 67 All: 59% males | All: 48 | Low back pain | DA: <1 wk: 64% PR: <1 wk: 100% | Limited data. Low participation rates & administrative errors which affected data. Study reports no significant differences in patient demographics (age & sex). Comparable clinical outcomes for both DA & PR |

*(Continued)*

**Table 1.** (Continued)

| First Author /Year of publication Country | Study Aim(s) | Study setting | Study Design | Eligibility criteria | Sample size (proportion of males) | mean age (SD) | Diagnosis (where specified) | Chronicity of Symptoms in weeks. Mean(SD) | Comments on fidelity / summary of study findings |
|---|---|---|---|---|---|---|---|---|---|
| Ludvigsson 2012 Sweden | to evaluate physiotherapist assessment and management of patients with musculoskeletal disorders in primary care, and to compare patient satisfaction with primary assessment by a physiotherapist or a GP. | Primary care | Sectional analysis of patient cohort | Adult patients seeking care for MSK disorders Excluded patients under 18 years. | DA: n = 51, 31% Males PR: n = 42, 57% males | DA: 46 (20) PR: 51(18) | ICD-10 diagnosis: Low back; Neck; shoulder; thoracic; knee; other | DA: < 4 wks: 18% 4-12wks 35% >12 wks 47% PR: < 4 weeks 24% 4-12 wks 26% >12 wks 50% | Data source: Medical records and follow up questionnaire for patients. Fidelity of triage was not explicitly reported but it was part of patient flow management where patients took up appointments as triaged by nurses. No significant differences in patient demographics for DA & PR |
| Ferguson et al 1999 UK | describe a self-referral service audit | Primary care | before & after service evaluation analysis. | Adult patients seeking care for pain symptoms <8 weeks' duration Excluded patients with Red-flag symptoms | All: 236 48% males | NR. | Non-specified, mostly MSK | All: <8 weeks | Data source: Service records No comparison data Retrospective audit mostly descriptive data Most common age group who self-referred: 30–50 years |
| Boissonnault 2010 USA | To explore successful implementation of a direct access physical therapy model at a large academic medical centre | Hospital/Rehabilitation | before & after service evaluation analysis. | Patients using direct access | All: 81 | NR | Spine and sports rehabilitation | NR | Pilot study No comparison data. Low service uptake but no adverse events or concerns about care. The analysis of patient data is descriptive gives overall rate of further referrals to Physician and further health utilisations |

*(Continued)*

**Table 1.** (Continued)

| First Author /Year of publication Country | Study Aim(s) | Study setting | Study Design | Eligibility criteria | Sample size (proportion of males) | mean age (SD) | Diagnosis (where specified) | Chronicity of Symptoms in weeks. Mean(SD) | Comments on fidelity / summary of study findings |
|---|---|---|---|---|---|---|---|---|---|
| Boissonnault 2016 USA | To investigate the extent of implementation and utilization of direct access to outpatient physical therapist services; identify barriers to and facilitators for provisioning of DA services, and; identify potential differences between facilities that do and do not provide DA services. | Hospital/ Rehabilitation | Survey | Directors of hospitals/ centres accessed through professional body. | NR | NR | Non-specified | NA | Iterative development of survey instruments with relevant stakeholders. 47 (52.8%) surveys completed. Participants had served in their current position for a mean of 9.3 years (range1–40). 41 were physical therapists by training, 5 occupational therapists, and 1 a certified athletic trainer. 20 (42.6%) of responders represented 25 hospitals/centers with DA 26 (55.3%) represented 36 hospitals/centers without direct access services, 1 (2.1%) in implementation process. Very low uptake of DA. |
| Chetty 2012a/ b UK | Describe results of diagnostic analysis and subsequent recommendations for implementation of nursing triage assessment in an occupational health and well-being service. | Occupational healthcare | Service evaluation audit | Nurses—working in occupational and wellbeing unit. Service users of occupational health and well-being service | Nursing (triage) staff (n = 7), Patient interviews (n = 22), | NR | Non-specified MSK | NR | Service data from documentary analysis, focus group and service user interviews evaluates a telephone triage service, exploring staff and users' perceptions. Face validity of study instruments was attempted prior to data collection. The subsequent survey of service users in this study does not examine the nurse triage but views on subsequent physiotherapy by DA/PR |

*(Continued)*

**Table 1.** (Continued)

| First Author /Year of publication Country | Study Aim(s) | Study setting | Study Design | Eligibility criteria | Sample size (proportion of males) | mean age (SD) | Diagnosis (where specified) | Chronicity of Symptoms in weeks. Mean(SD) | Comments on fidelity / summary of study findings |
|---|---|---|---|---|---|---|---|---|---|
| Mant et al 2017 UK | To explore GPs level of satisfaction, their opinions of current NHS physio direct service and any suggestions for future improvements | Outpatient | Survey | GPs within the specified service area | All: 104 | NR | Non-specified MSK | NR | A purposive sampling but low response rate 33%. Possible increase in non-response bias of GPs with less than 5 years' experience in the area and therefore no knowledge of the service. |
| Harland et al 2016 UK | To explore the attitudes of stakeholders (clinical- GPs & Physios) regarding DA services. | Mixed (mostly Primary Care) | Survey | GPs or Physio working in services with or without DA. | All: n = 541 PTs: 488, 18% males GPs: 68, 43% males. | NR | Non-specified MSK | NR | Sampling/ recruitment from known networks and email cascade. May not be representative. Low GPs response rate. Possible responder bias with those with strong views and those with access to PD services more likely to respond. |
| McCallum 2012 US | To describe factors that affect direct access physical therapist practice. | Mixed (mostly primary care) | Survey | Licenced and registered physiotherapists in the state. | All: 1266, 25% males | NR | Non-specified MSK | NR | Survey instrument developed with clinician focus group. 31.0% of responders practiced DA, were mostly females. No significant differences in age range across DA & PR Physiotherapists. PTs in DA group were more experienced (23.6% had > 25 years) and had more advanced degrees. |
| Pearson et al 2016 UK | To describe patient acceptability and experience of the PhysioDirect service compared to usual PR care | Primary care | Qualitative Interviews | Inclusion in a previous telephone triage trial–Physio Direct. | All: n = 57, 46% males. | 58(16.88) | General MSK–back, upper & lower limb, and multiple areas of pain | NR | Good qualitative methodology. Sample reflected wider range of service users. Usual care views also collected to gain direct comparison. |

DA: Direct Access, PR: Physician referred, NR: not reported

*DA: true self-referral + GP suggested self-referral, MSK: musculoskeletal

≠: Queried true randomisation process

1,988 clinicians (including GPs, Physiotherapists, nurses and other allied healthcare professionals) regarding direct access, self-referral and/or triage services in the management of patients with MSK conditions [17, 23, 31, 33, 35, 36].

Across the nine studies which presented direct comparison data (in total 25,122 patients with experience of direct access services versus 56,992 patients who had been managed through usual GP-led services), patient characteristics were reported not to be statistically significantly different with reference to age and gender. However, those who accessed direct access services in nine studies were on average more often female, younger and slightly more educated [14, 16, 17, 25, 28, 30, 32, 38, 40]. Out of eight studies which presented data on the chronicity of patient symptoms [15, 17, 25, 28, 30–32, 40], only three reported differences between groups [17, 24, 30]. Direct access patients were slightly more likely to present with less chronic (i.e., shorter mean duration of) symptoms up until the time of being seen by a physiotherapist (e.g., Mallet et al mean number of days for direct access 3.55, ±2.7, vs 30.99, ±15.4 for GP-led care [24], and; Holdsworth et al where up to 51% of direct access patients were seen in less than 6 weeks versus 23% of patients receiving GP-led care) [17]. However, Denninger et al found patients using direct access services slightly more often had a chronic presentation (63% versus 54%) [30].

## MSK triage and direct access service models in primary care settings and associated barriers/ facilitators—study objective 2

MSK triage or direct access services across included studies, were classified into three main groups, based on their distinctive features about how direct access was operationalised (refer to S5 Table for further details):

- *open access* where patients by request (telephone, walk-in, self-referral form) gain direct access to non-medical practitioner (e.g. physiotherapists).

- *combination models* which often combines open direct access to non-medical practitioners with a triage process to assess patient suitability, or ensures on site access to GPs for review and input on a needs basis.

- *service based pathways* which are essentially non-patient level interventions. Patients were free to choose GP-led care even when access to non-medical practitioners was available in the service. Direct access was usually by face-to-face open access.

**Open access models.** The 15 studies mostly involved GP practices where direct access services were advertised directly to patients, who were free to access non-GP care directly (mostly physiotherapy) for the management of their MSK conditions [14, 15, 17, 20, 21, 25–28, 30, 31, 34, 37–39]. Furthermore, care facility staff (reception personnel, nurses, and physician assistants) not involved in provided MSK care, but who may field patient calls, were usually trained and encouraged to present direct access options to patients where appropriate. Within this model, there were often no strict requirements or set criteria for triaging MSK patients for physiotherapy assessments and management.

*Barriers & facilitators associated with open access models.* Actual barriers to accessing care for MSK conditions were less frequently experienced (or mentioned) in open access models. Perceived barriers (mainly from health care professionals' perspectives), were however reported and mostly related to patient safety. Medical professionals were concerned about physiotherapist's competence in medical screening and differential diagnosis and subsequent, overall increase in resource utilisation (e.g., imaging, medications, McGill et al. [39]). Other

concerns were a negative effect on doctor-patient relationships (e.g., "fear of de-skilling of GPs" and patient picking up GP's lack of specific MSK skills) [31], and problems with acceptability to patients (e.g., cultural requirement for GP diagnosis prior to physiotherapy referral) [31].

In terms of organisational issues, barriers associated with implementing open access services included: lack of health care provider or administrator knowledge regarding outpatient direct access and its legality, robustness and provision of risk management policies, facility-specific requirements and training for physiotherapists offering direct access services, organisation's scheduling system problems, decreased reimbursements or denied payments for patients receiving outpatient physiotherapy via direct access, increased time demands on the physiotherapy services, concerns regarding physiotherapy scope of practice, increased costs of professional liability insurance, and overutilisation of physiotherapy services [15, 21, 31, 34, 39].

Overall, in comparative study designs, healthcare facilities offering this model of care were less likely to perceive listed factors as insurmountable barriers to management of MSK patients through direct assess compared to organisations which did not offer these services [17, 20, 28]. To enhance care and service delivery, these studies often suggested adequate training of direct access providers, high quality administrative support and patient awareness as possible solutions to overcoming associated barriers. Furthermore, timely and efficient access to physiotherapy, and enhanced patient satisfaction with care were reported to facilitate implementation of direct access in those facilities that offered this model [17, 20, 21, 26, 28, 34].

**Combination models.** Of ten studies classified as combination models of direct access, six [19, 22–24, 29, 32, 40] report observational data (from two cohorts [29, 32]; three service evaluation audits [19, 22–24] and one cross-sectional analysis of health records data [40]). The remaining four focused on exploration of views regarding direct access/ self-referral services as perceived by patients, practitioners and the general public [18, 33, 35, 36].

The studies employed hybrid features of open access using both telephone-based or face-to-face delivery of patient assessment and initial management. Typically, the combination model included an extra layer of filtering where patients seeking care for MSK conditions through self-referral were often triaged through telephone contact by specially trained physiotherapists or other personnel to the most appropriate care available for their condition including direct access to physiotherapy for self-management advice or GP assessment followed by physiotherapy referral where appropriate [22, 23, 32, 40]. In addition to telephone contact, triaging was also sometimes performed face-to-face when patients make contact with such health care facilities. Triage systems usually followed locally developed protocols or algorithms, and were varied. In addition, to address concerns regarding safety, some of these services required the presence of onsite physicians who may be asked to review patients (where necessary), in order to mitigate against risks of red flags and missed diagnoses [32, 40].

*Barriers & facilitators associated with combined access models.* The combined model of access to direct services/self-referral options included further administrative procedures typically initiated at telephone contact from the patient via a telephone triage appointment, then followed by face-to-face consultations [32, 33, 35]. There were also uncertainties about the proportion of patient caseload likely to be adequately addressed through phone consultation, thus preventing further face-to-face consultations and healthcare costs [33, 35]. A number of studies which engaged the combination models were found to have described fidelity of planned access to care through self-referral options but actual delivery did not always appear to have been implemented according to plan [24, 32, 40].

Within the combined model, especially where patient care had not progressed further to actual face-to-face physiotherapy or GP assessment and follow-on care and patients were advised by telephone to self-manage, patients reported perceptions of inadequacy of triage staff in addressing the presenting MSK problem, lack of insight into the impact of the MSK

problem on patients' health and wellbeing as well, as unmet expectations regarding management of the MSK problem [22, 23]. However, these barriers were not reflected by patients who were triaged to at least one or more physiotherapy sessions with or without further GP consultations [22, 23, 29, 32].

**The service-based pathway model.**   The only study in this model was a cluster (pilot) trial which featured service level comparisons of outcomes of direct access for MSK and involved multiple professionals [16]. This study did not compare patients receiving direct access with those who received usual GP-led care, but compared GP practices where an open direct access pathway was available to patients with MSK conditions with practices where it was not. As a result, not all patients in the intervention arm (where direct access to physiotherapy was available) accessed direct access services.

*Barriers & facilitators associated with the service based pathway model.* There was limited evidence to fully explore and profile this model of access. There was an observed increase in the number of overall referrals to physiotherapy in intervention practices (offering open direct access services) compared with service-level data collected in the year prior to this pilot trial, but the authors attributed this, in part to the active marketing of the direct access pathway during the trial. The authors envisaged a possible need for staff training, organisational set-up, procedures and advertisement of the services, which may be required to fully implement this service based model [16].

## Patient related outcomes of MSK triage and direct access services—study objective 3

**Clinical outcomes (pain and disability).**   The *evidence base* for the outcome of MSK triage and direct access services on patient pain and functional disability included nine studies [15–17, 24, 27–30, 32], of which six offered open access service models to patients with MSK conditions. A wide range of patient reported measures were used for assessing pain across these studies and included visual analogue scales [17, 24, 29, 32], percentage decrease in pain [28], numerical pain rating scales [30], Pain Self-Efficacy Questionnaire [27], Back pain checklist [15] and global assessment of change [16]. Similarly, functional limitations were assessed by Patient Specific Functional Scale [27], Oswestry Disability Index [30], Sickness Impact Profile and the physical component summary measure from the SF- 36v2 questionnaire [15, 16].

**Outcomes/Magnitude of effects.**   Seven studies (six of which were open access models) reported data on pain and functional outcomes for patients who assessed MSK care via direct access compared to GP-led care [15–17, 24, 28, 30, 32]. Across these studies, differences in group means were consistently small and statistically insignificant (e.g.78% for self-referral vs. 80% for GP-led-care in Overman et al. [15]; 7.2% of direct access vs. 7.6% of GP-led care patients reported complete recovery from symptoms at 12 months in Bishop et al. [16]). An exception to this trend was found in one study which was a combination model type of direct access, and reported that pain and functional outcomes in the short term (up to 3 months) were slightly better for MSK patients who were managed by usual GP-led care compared to direct access services [32].

**Bottom line.**   In the long term, improvements in pain and functional disability were consistently similar between direct access patients and GP-led care groups.

**Clinical outcomes (QoL).**   *The evidence base* consists of five studies: two combination type service models [29, 32], two open access type models [30, 31] and a service based pathway model [16]) studied and assessed patients' quality of life following direct access consultations. All used a validated quality of life questionnaire, such as the EQ-5D, SF-12 or 36 mental/physical component scores.

**Outcomes/Magnitude of effects.**   Of the five studies, two were cohort studies with no control /comparison group, hence data analysis was in comparison to baseline [29, 31]. The study by Deninger et al., a comparative cohort reported no quantitative outcome data for QoL subsequent to baseline [30]. The study however, found similar (no significantly different) improvements in patients' quality of life irrespective of direct access to physiotherapy services or GP-led-care for up to two years after initial consultations. Similarly, Bishop et al. reported similar improvements in QoL for MSK patients who accessed GP-led care and direct access service pathways [16]. On the other hand, Ludvigsson et al., a comparative cohort study showed that participants who accessed care for their MSK conditions via direct access services reported better quality of life at 3 months post initial consultation (mean EQ 5D (standard deviation SD) 0.65 (0.22) for direct access groups vs. 0.51 (0.30) for GP-led care, p = 0.014) [32].

**Bottom line.**   Similar to pain and functional disability outcomes, improvements in patient health related quality of life were comparable between direct access patients and GP-led care groups. As study design and outcomes of care were mixed, the effect of particular model/type of services by which patients accessed MSK triage and direct access to physiotherapy on overall quality of life is unclear.

**Safety outcomes (adverse effects and missed red-flag diagnoses).**   *The evidence base* consists of five studies which specified serious adverse events or missed red-flag diagnoses as an outcome for their study. All were open access type/models [20, 26, 30, 31], with the exception of the only service pathway type/model of access [16].

*Outcomes/Magnitude of effects*. Of the five studies, only two were comparative in design, and reported no adverse events by GPs or physiotherapists [16, 30]. The review of medical records in the trial by Bishop et al also identified no evidence of missed serious pathology in MSK patients who received care through direct access [16]. Similarly, across the three other studies evaluating outcomes after introduction of direct access services, there was no record of any adverse event related to patient management through direct access, nor were there reports of physiotherapists involved in litigation or disciplinary action pertaining to the examination and treatment of patients seen through direct access [20, 26, 31]. There was also no report of missed diagnosis or delay in diagnosis of MSK conditions as a result of accessing care through MSK triage and direct access in these studies. In the trial by Overman et al.[14] adverse events or safety issues was not a specified outcome, but were reported as part of routine data [15]. However, three patients were noted with red flag conditions (unrelated to the MSK problem) which were not immediately spotted by physiotherapists but this did not result in adverse outcomes as the therapists (at initiation of treatment /management) did refer these patients back to physicians who then diagnosed and put in place appropriate management plan for these patients.

*Bottom line*. Results from the five studies do not provide evidence of worse outcomes, adverse effects, or missed red-flag diagnoses for patients with MSK conditions who access care through MSK triage and direct access (irrespective of the type/model of access). An overall absence of evidence of harm as a result of direct access to physiotherapy services was found but the available studies were not designed to robustly assess this.

## Socio-economic outcomes (work absence and sickness certification)

*Evidence base*. Five of the included studies (two open access type/models [17, 39], two combination type/models [29, 40], and one service based pathway [16] provided data and contributed to evidence regarding work absence and sickness certification for MSK patients who accessed care via direct access to physiotherapy.

*Outcomes/Magnitude of effects*. Defined mostly as the number of days of work absence as a result of pain, three of the studies [16, 17, 39], found that, proportions of work-related absence

due to MSK pain differed significantly in favour of those who had direct access to physiotherapy services compared to usual GP-led care. For example, Holdsworth et al. reported mean MSK related work absence (days ±SD) as 2.5, ±10.6, for self-referrers compared with 6.0, ±19.6 for GP-led care group) [17]. The study by Bishop et al. found the proportion of patients who reported having taken time off work as a result of their MSK condition over 12 months was similar across both control and intervention practices who had access to the open direct access pathway [16]. However, further analysis based on the cost of absence from work due to MSK condition showed that patients who had access to MSK triage/ direct access pathways required fewer self-reported days off work, and overall lower costs of work related loss at 12 months (mean difference in work related loss due to MSK was up to £200.00).

Bornhoft et al. [40] defined socioeconomic outcomes in terms of sickness certification, i.e., the proportion of patients who received doctors' notes for sick-leave for MSK related problems, and also found that patients who had direct access to physiotherapy services were overall less likely to be in receipt of sickness certification from GPs (odds ratio with 95% confidence interval 0.55 (0.42–0.71) at 6 months and at 12 months 0.58 (0.44–0.77); p <0.001).

*Bottom line*. Evidence from four comparative studies consistently shows that patients with MSK conditions who access care through MSK triage and direct access (regardless of access types/model) report less work-related absence and sick leave episodes as a result of their MSK conditions compared to those receiving usual GP-led care.

## Health care utilisation (costs, further consultations, prescriptions, tests, referrals, and impact on GP workload/ services)

*Evidence base* includes 15 studies which reported health care utilisation outcomes. Of these, 11 are open access type/model services [14, 17, 25, 27–31, 37–39], four studies provide evidence for combination type/model services [24, 29, 32, 34], and a final one concerned a service pathway model [16]. A wide range of definitions and measures were used to assess healthcare utilisation outcomes, but were mostly in terms of changes in GP workload (initial and further consultations), additional tests and referrals, and cost of care following implementation of direct access for MSK pain.

*Outcomes/Magnitude of effects*. Though estimations of the total cost of care (and/or reimbursed amounts in case of insurance claims data) varied across studies, evidence from five studies with comparative designs found overall healthcare costs to be lower on average by 10–20% for direct access s compared to usual GP-led care for MSK [24, 25, 28, 30, 32]. For example, Badke et al. reported the mean total cost of care per patient (SD) for direct access patients as $2423.5 (2555.3) compared to $3878.7 (2923.8) for GP-led care [28]. Denninger et al. also reported total cost care per patient (SD): 1542 (108, 2976) for direct access versus 3085 (1939, 4224) for GP-led care [30]. In the same vein, observed patterns for analgesics and NSAIDs prescriptions were mostly less for direct access / self-referral services but sometimes comparable to GP-led usual care across studies (e.g. Boissonault [21, 34], McGill et al. [39]: Medication use: 24% for direct access compared to 90% for GP-led care while radiology use was 11% for direct access compared to 82% for GP-led care; analgesics use and muscle relaxants was 10% for direct access patients compared to 42% for GP-led care—Overmann et al. [15]. Furthermore, the number of referrals (>1) to a specialist or further consultation for the same disorder for up to 1 year following index consultations was between 2% (Holdsworth et al. [17]- a trial) and 10% lower (Bornhoft et al. [40]—a cross sectional comparison of patient groups), compared to usual care.

*Bottom line*. Consistently, evidence from 10 studies with comparative designs shows that usual GP-led care for patients with MSK conditions are associated with relatively higher

health-care utilisation and costs compared to provisions for any model of MSK triage direct access options.

Table 2 presents a summary of findings for the different patient related outcomes across the three models of MSK triage and direct access services.

## Discussion

This systematic review has systematically identified, synthesised and graded available evidence regarding outcomes of MSK triage and direct access in primary/community care, non-GP-led, services considering patient outcomes (pain, disability, work absence and sickness certification), safety (e.g. missed red-flag diagnoses), socio-economic and health care costs (consultations, prescriptions, tests, referrals, and impact on GP workload/services). The different models of direct access services, as well as the barriers and facilitating factors associated with the implementation of these services were also profiled. The aims of this review are important in terms of understanding if non-GP first models of care are relieving GPs of existing workload rather than creating supplier induced demand. The other objective about mapping and understanding current practice, helps to ascertain if homogenous models are being used or if heterogeneity makes broad comparisons of outcomes difficult for the purpose of commissioning of care.

Across a wide array of primary/community care settings included in this review, patients who had experienced, or chose to access care for their MSK conditions through direct access to physiotherapy services, varied from study to study but were not significantly different to those who had been managed through usual physician referred or GP-led services. This was found to be generally true with reference to age, sex and duration of symptoms. However, those who accessed direct access and self-referral services were often younger, slightly more educated and having better socio-economic status. Apart from the well-known effect of education and socio-economic status on health access and health disparity, the slight differences in the profile of patients availing themselves of the opportunity to self-refer directly to physiotherapy services may also be as a result of how access to these direct access services were advertised [16, 17], organised [21, 25, 34], and implemented [16, 21, 25, 34]. It may be that targeted education and advertisement to underserved groups or population sub-groups might be required for widespread implementation.

In this review, an attempt has been made to understand the nature of the wide array of direct access services for MSK patients as well as to classify this. Approximately 60% of available evidence (n = 15 studies) align with open access models and appear to be most accessible to patients compared with combined models of care which often feature an extra layer of triaging and procedural complexities in the management of patient flow through these services. The increased time and monetary costs associated with the extra layer of patient filtering may make the combination model less desirable compared the open access models. Understandably, many of the combination models of care were set up to mitigate risks to patients and also ensure that physiotherapy services are rightly accessed only by those who need it. Furthermore, within combination models, there is the possibility that younger, patients with less chronic symptoms and co-morbidities were often triaged for education and advice for self-management through telephone consultation while older patients with "more complex physical health" needs may have been filtered, first for GP assessment and subsequent physiotherapy referral as appropriate. However, there was no empirical evidence to support this assumption as none of the included studies except for Bishop et al. [16] evaluated direct access options at service based levels.

In terms of patient oriented and clinical outcomes of care such as pain, and functional disability, the outcomes of direct access models did not show large or significant differences

**Table 2. Summary of findings.**

| Treatment Options | Evidence treatment options across regional musculoskeletal pain presentations | | | | |
|---|---|---|---|---|---|
| | Service Model | Evidence base | Outcomes / Effects | Comments | Overall Strength of evidence (Grade) |
| *Clinical outcome (pain and disability)* | Open access | 1 Trial (Holdsworth 2007, Overman et al 1988), 2 Cohorts (Badke et al 2014, Denninger 2018); 2 Service evaluations (Ojha 2015, Mallet 2014). | Small differences between groups (e.g., Mean functional improvement score at discharge 15.2 ±11.7 for self-referred patients vs 14.6 ±10.6 for GP led care; p = 0.77) on a 0–100 scale for function) and (e.g., percent decrease in pain 64.6% for self-referrers vs 66.6% for Physician referred patients; p = 0.76), Badke et al. 2014; Mean improvement in function from baseline, 54%; 95% CI: 46%, 62%) and pain (mean difference, 4 points; 95% CI: 1, 7 points), with no differences between groups (P>.05), Denninger 2018). | Overall, patients displayed good clinical improvement in disability and pain, with no differences between groups (P >.05). Between group differences in pain and function were also not sustained in the long term (>12 months). | ** **Limited evidence** |
| | Combination | 2 cross-sectional analysis of patient cohort. Ludvigsson 2012; Phillips 2012 | Mean (SD) summary index (EQ VAS) of self-rated health including pain and functional disability on a scale from 0 to 100: 67 (18) for self-referred patients vs. 56 (19) for GP-led care; p = 0.006). Ludvigsson 2012. Mean pain intensity (VAS (SD)) 6.91 (9.4), p<0.001 at 3 months follow up. | Significant differences were found between groups. Relatively small data-set (n = 93) from a patient cohort. Philips et al 2012 was compared to baseline but did not include comparison group data. | |
| | Service based pathway | 1 cluster randomised trial. Bishop et al 2017 | Perceived change from baseline:4% of self-referred patients vs. 6.5% of GP-led care patients reported complete recovery at 6 months | Evidence from pilot trial. (cluster randomisation based on GP practices). | |
| *Clinical outcome (Quality of life)* | Open access | 2 Cohort (Denninger 2018; Goodwin 2016/Moffatt 2017). | Beneficial effects demonstrated. Small, statistically insignificant differences between groups at follow-up (e.g. percent change in pre-post EQ 5D mean (SD) at 6 months 0.13 (0.27) Goodwin 2016). | Comparable improvements (slightly better among self-refers) in QoL outcomes for up to 2 years across studies. | ** **Limited evidence** |
| | Combination | 2 cross-sectional analysis of patient cohort. Ludvigsson 2012; Phillips 2012 | e.g., mean EQ 5D (SD) 0.65 (0.22) for self-referred groups vs. 0.51 (0.30) for GP led care at 3 months, p = 0.014 Ludvigsson et al; and 0.82 (0.2) at 3months, p<0.001 Phillips et al 2012. | Unadjusted analysis | |
| | Service based pathway | 1 cluster randomised trial. Bishop et al 2017 | Mean EQ 5D score (SD) for control practices vs intervention practices respectively: @ baseline: 0.565 (0.246) vs. 0.544 (0.262) @ 6 months 0.602 (0.251) vs. 0.594 (0.262) @ 12 months 0.615 (0.254) vs. 0.606 (0.258) | Quality of life increased similarly in both arms compared to baseline across all follow-up time points | |

*(Continued)*

**Table 2.** (Continued)

| Treatment Options | Service Model | Evidence base | Outcomes / Effects | Comments | Overall Strength of evidence (Grade) |
|---|---|---|---|---|---|
| | | **Evidence treatment options across regional musculoskeletal pain presentations** | | | |
| *Safety outcomes (adverse effects and missed red-flag diagnoses)* | Open access | 2 Cohort (Denninger 2018, Goodwin 2016) 2 service evaluation (Mintken 2015, Moore 2005). Other studies without safety as a priori outcomes: (McGill et al 2013, Ojha 2015, Pendergast et al 2012, Holdsworth 2007, Greenfield 1975, Boissonnault 2010, 2016, Desjardins-Charbonneau et al 2016) | No adverse events/effects, missed red flag diagnoses due to accessing care through MSK triage and direct access/self-referral across all included studies. | MSK triage/direct access presented no higher risks to patients. However, most services included specially trained and/or more senior professionals. | *** **Moderate evidence** |
| | Combination | Other studies without safety as a priori outcomes: Ferguson et al 1999 | | Informal liaison with GPs, access to patient medical notes, and use of pre-defined protocol/ checklists for minimising mis-diagnosis. | |
| | Service based pathway | 1 cluster randomised trial. Bishop et al 2017 | | No evidence that the direct access pathway led to adverse events, missed diagnosis of serious pathologies. No comparison with control practices without direct access services. | |
| *Socio-economic outcomes (work absence and sickness certification)* | Open access | 1 Trial (Holdsworth 2007) 1 cross-sectional analysis (McGill et al 2013) | (Mean MSK related work absence, S.D., range (days): 2.5, ±10.6, 0 to 120 for self-referrers; vs. 6.0, ±19.6, 0 to 300; p = 0.048). Holdsworth et al 2007 94% drop in lost time from work due to MSK related condition over 12 months. | Consistently large differences in favour of direct across/self-referral for up to 12 months across studies. | *** **Moderate evidence** |
| | Combination | 1 cross sectional analysis (Bornhoft 2015) 1 analysis of patient cohort (Phillips 2012). | N (%) of sick-leave recommendations for direct access and GP led care respectively. 82 (14.1%) vs. 369 (23.2%) @ 6months 73 (15.1%) vs. 338 (23.5%) @ 12 months. Bornhft 2015. Mean (SD) Sickness absence @ baseline and @ 3months 4.6 (12.6) vs. 1.45 (9.7); p <0.05 Mean (SD) Work performance @ baseline and @ 3months 75.9 (19.6) vs. 87.8 (13.2); p <0.001. Phillips et al 2012 | Significant differences in work related outcomes relative to baseline. | |
| | Service based pathway | 1 cluster randomised trial. Bishop et al 2017 | Mean (SD) work related costs associated with MSK conditions: £740.30 (2084.75) for control practices vs £ 539.36 (2069.43) for intervention practices who accessed care via MSK triage/ direct self-referrals. | Work related absence costs were significantly higher for patients without direct access. Outcome over 12 month period. | |

(*Continued*)

**Table 2.** (Continued)

| Treatment Options | Service Model | Evidence base | Outcomes / Effects | Comments | Overall Strength of evidence (Grade) |
|---|---|---|---|---|---|
| | | **Evidence treatment options across regional musculoskeletal pain presentations** | | | |
| *Health care utilisation (costs, further consultations, prescriptions, tests, referrals, and impact on GP workload/services)* | Open access | 2 Trial (Holdsworth 2007, Greenfield 1975), 3 Cohorts (Badke et al 2014, Denninger 2018, Goodwin 2016); 2 Service evaluations (Ojha 2015, Swinkels 2014). 4 cross-sectional analysis (McGill et al 2013, Mitchell et al 1997, Pendergast et al 2012) | Badke- Mean total cost of care per patient (SD): $2423.5 (2555.3). Mean total cost of care per patient (SD): $3878.7 (2923.8) Denninger 2014. Total cost care per patient (SD): 1542 (108, 2976). For DA vs 3085 (1939, 4224) McGill et al 2013: Medication use: Medication use: 24.07% for DA compared to 90.53% for GP led care. Radiology use: 11.11% compared to 82.11% for GP led care. | Overall, consistently significant differences in health care utilisation costs (higher for usual GP-led care compared to MSK triage and direct access/ self-referral) | ***Moderate evidence |
| | Combination | 1 cross sectional analysis (Bornhoft 2015) 2 analysis of patient cohort (Phillips 2012; Ludvigsson 2012. 1 service evaluation (Mallet 2014) | | | |
| | Service based pathway | 1 cluster randomised trial. Bishop et al 2017 | | | |

*Very weak evidence: Perspective / opinions only/ Absence of empirical data (from qualitative or quantitative studies).

** Limited evidence: Some empirical evidence from cohort and cross-sectional observational studies, lacking comparisons with usual GP led care, AND when there were small, inconsistent, or non-significant differences in patient related outcomes, OR without.

*** Moderate evidence: Some empirical evidence from trials, good quality cohort and cross-sectional analyses of large data sets including, comparisons with usual GP led care, and /or with small to moderate but consistent effects on patient related outcomes.

**** Strong evidence: Evidence from good quality trials, cohort and cross-sectional analyses of large data sets including direct access, comparisons with usual GP led care, and /or with moderate to strong consistent effects on patient related outcomes.

compared to those observed from GP-led models of care, neither did outcomes differ significantly between the different models of direct access services. Also our findings clearly show no evidence for increased risk associated with assessing care for MSK symptoms through any of the direct access models to physiotherapy services, however, incidence of adverse outcomes was small (not surprisingly) in this group of patients, and many of the included studies were not designed to assess these, or were simply not sufficiently powered to detect differences in risk.

What was most obvious was the difference in healthcare utilisation, costs and socioeconomic outcomes between direct access and GP-led care. The caveat to this is that the earlier reported differences of patients being younger and having higher socio-economic status could impact health care utilisation, work outcomes and subsequently costs. More importantly, methods of estimation of total costs of care varied between studies and many of these direct access models of care (especially the combination models) also required GPs to be present on site for consultation as needed, but the burden of these aspects of care were not usually accounted for.

The barriers and facilitators associated with the three models of care profiled in this study largely reflects organisational and administrative issues and we feel this is an important finding in this manuscript. Often, research is undertaken with a primary focus of informing clinical practice rather than taking an organisation and systems based approach to rethinking models

of care. It may be that ineffective healthcare delivery is not always as a result of bad science or the proficiency of healthcare professionals, but due to organisational or administrative reasons. The barriers and facilitators found in this review suggest that new evidence-based approaches to accessing care is needed. Given the economic differences in cost of care and minimal gains in clinical outcomes as a result of direct access to MSK, large gains in patient oriented clinical outcomes can be gained as a result of simple cost effective solutions relating to the administration and organisation of care.

## Strengths and limitations of the review

This review provides a summary of available evidence regarding the outcomes of triage and direct access services for the management of MSK conditions in primary/community care, drawing together findings from a variety of evidence sources from across the world. Further strengths of this review include a comprehensive search strategy and a mixed methods synthesis process to capture all available information on this topic.

There are also limitations to this review. The evidence synthesis was challenged by the mixed sources of primary data including observational, uncontrolled and mostly non-randomised studies, use of different methods for data collection and a wide range of outcomes. Data were therefore not suitable to conduct a statistical pooling (meta-analysis) of outcome data. In addition to the wide heterogeneity of design and available data, many of the included studies showed methodological limitations, precluding any strong statements regarding the effects of direct access MSK services. We therefore took a cautious approach to the assessment, synthesis, grading, and interpretation of the available evidence. Specifically, due to the amount and type of evidence presented by the studies in this review, the modified GRADE assessments as used in the present study is not be directly comparable to standard GRADE assessments and must be interpreted with caution.

## Implications for future practice, health care planning and research

There is a very wide variation in currently available direct access services for MSK and the existing state of evidence is poor. Within the literature, services were often very poorly described and it is difficult to unpick how direct access services were operationalised or implemented. Many of the existing direct access models required doctors to be present and are as such not a replacement to GP care but adjunct in those cases. With the current surge in policies driving implementation of non-medical direct access for patients with MSK conditions, is also the risk of implementing suboptimal care due to poor description of services and lack of high-quality research with suitable, bias free comparisons.

Many of the included studies were not designed or adequately powered to evaluate equivalence or non-inferiority among the different modes of access to care for MSK conditions. However, outcomes of care and safety were consistently similar across these studies, although it must be noted that available studies were not designed to robustly assess potential harm or adverse outcomes from the introduction of direct access. Though small and similarly not powered to examine equivalence of GP-led care over direct access for MSK patients, a recent trial also finds no significant differences in pain, and functional disability [41]. The services proposed here therefore seem to be a more efficient and less costly service model for patients with MSK conditions and/or have potential to help reduce GP workload. Undeniably, direct access MSK services are novel and have potential to transform current care for patients with MSK conditions in a positive manner. Careful consideration must be given to putting in place evidence-based support systems and resources (suitably trained staff) that will assess for and ensure sustainability, safety and optimum care for MSK patients.

## Conclusions

Available evidence to date suggests that, socio-economic (health care costs, utilisation, and work absence) outcomes may be better, and there is no difference between clinical (pain, function, safety) outcomes for patients with MSK who accessed care through non-medical direct access services compared to those who access care through usual GP-led services. As a result, many patients seeking primary/community health care for MSK conditions, and who would usually be assessed and managed by in GP-led services could be adequately assessed and managed through direct access to physiotherapy services. However, due to the paucity of strong empirical data from methodologically robust studies, a scale up and widespread roll out of non-medical direct access services can, as yet, not be assumed to result in long term health and socio-economic gains without careful considerations of the elements and the most appropriate access model to be implemented in each care setting. This will ideally be tested by evaluating the full range of relevant patient and resource outcomes between different service based pathways in order to optimise care for patients with MSK pain.

## Supporting information

**S1 Checklist.**
(DOC)

**S1 Table. Detailed search strategy (medline).**
(DOCX)

**S2 Table. Detailed eligibility criteria.**
(DOCX)

**S3 Table. Modified GRADE criteria.**
(DOCX)

**S4 Table. Methodological quality appraisal MMAT tool.**
(DOCX)

**S5 Table. Classification of direct access and first contact service models.**
(DOCX)

## Acknowledgments

The authors acknowledge members of the patient involvement and engagement—RUG who contributed to consultations regarding the research questions, design and conduct of this study.

## Author Contributions

**Conceptualization:** Opeyemi O. Babatunde, Annette Bishop, Elizabeth Cottrell, Alyson L. Huntley, Danielle A. van der Windt.

**Data curation:** Opeyemi O. Babatunde, Annette Bishop, Elizabeth Cottrell, Joanne L. Jordan, Nadia Corp, Katrina Humphries, Tina Hadley-Barrows, Danielle A. van der Windt.

**Formal analysis:** Opeyemi O. Babatunde, Annette Bishop, Elizabeth Cottrell, Joanne L. Jordan, Katrina Humphries, Tina Hadley-Barrows, Alyson L. Huntley, Danielle A. van der Windt.

**Funding acquisition:** Opeyemi O. Babatunde, Annette Bishop, Danielle A. van der Windt.

**Investigation:** Opeyemi O. Babatunde, Annette Bishop, Elizabeth Cottrell, Nadia Corp, Katrina Humphries, Tina Hadley-Barrows, Alyson L. Huntley, Danielle A. van der Windt.

**Methodology:** Opeyemi O. Babatunde, Joanne L. Jordan, Nadia Corp, Alyson L. Huntley, Danielle A. van der Windt.

**Project administration:** Opeyemi O. Babatunde, Danielle A. van der Windt.

**Resources:** Opeyemi O. Babatunde.

**Supervision:** Danielle A. van der Windt.

**Validation:** Opeyemi O. Babatunde, Annette Bishop, Elizabeth Cottrell, Joanne L. Jordan, Katrina Humphries.

**Visualization:** Opeyemi O. Babatunde.

**Writing – original draft:** Opeyemi O. Babatunde.

**Writing – review & editing:** Opeyemi O. Babatunde, Annette Bishop, Elizabeth Cottrell, Joanne L. Jordan, Nadia Corp, Katrina Humphries, Tina Hadley-Barrows, Alyson L. Huntley, Danielle A. van der Windt.

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
