## [Decision Letter · Decision Letter 0]

21 May 2020

PONE-D-20-05712

A Systematic Review and Evidence Synthesis of Non-Medical Triage, Self-Referral and Direct Access Services for Patients with Musculoskeletal Pain.

PLOS ONE

Dear Dr. Babatunde,

Thank you for submitting your manuscript to PLOS ONE. After careful consideration, we feel that it has merit but does not fully meet PLOS ONE’s publication criteria as it currently stands. Therefore, we invite you to submit a revised version of the manuscript that addresses the points raised during the review process.

We would appreciate receiving your revised manuscript by Jul 05 2020 11:59PM. To enhance the reproducibility of your results, we recommend that if applicable you deposit your laboratory protocols in protocols.io, where a protocol can be assigned its own identifier (DOI) such that it can be cited independently in the future. For instructions see: http://journals.plos.org/plosone/s/submission-guidelines#loc-laboratory-protocols

We look forward to receiving your revised manuscript.

Kind regards,

Alison Rushton

Academic Editor

PLOS ONE

Journal Requirements:

2. Please consider inclusion of large tables as supplementary information files.

3. Please clarify in the Methods section that more recent articles published since February 2018 have been considered in your review.

Additional Editor Comments (if provided):

Please reflect on the comments from reviewers (particularly reviewer 2) to see if any amendments are warranted to further enhance the quality of the manuscript.

Reviewers' comments:

Reviewer's Responses to Questions

**Comments to the Author**

1. Is the manuscript technically sound, and do the data support the conclusions?

Reviewer #1: Yes

Reviewer #2: Yes

Reviewer #3: Yes

2. Has the statistical analysis been performed appropriately and rigorously? 

Reviewer #1: N/A

Reviewer #2: Yes

Reviewer #3: Yes

3. Have the authors made all data underlying the findings in their manuscript fully available?

Reviewer #1: Yes

Reviewer #2: Yes

Reviewer #3: Yes

4. Is the manuscript presented in an intelligible fashion and written in standard English?

Reviewer #1: Yes

Reviewer #2: Yes

Reviewer #3: Yes

5. Review Comments to the Author

Reviewer #1: PONE-D-20-05712

Manuscript title: A systematic review and evidence synthesis of non-medical triage, self-referral and direct access services for patients with musculoskeletal pain

Comment:

This article researches the evidence regarding characteristics, outcomes, barriers nad facilitators of musculoskeletal (MSK) triage and direct access services using meta-analysis technique. This view point is nice and collected data was huge volume. The results demonstrated that methods and outcomes of care did not much difference between GP-led care and direct access. However, studies were so heterogeneous that statistical analysis was not fulfilled in the article. I think this article is worth publishing in Plos One because much information is in the article.

One thing authors should put full spelling of MSK before first appearance in ABSTRACT.

Reviewer #2: This paper reports the existing evidence regarding characteristics, outcomes, barriers and facilitators of MSK triage and direct access services. There is also a substantial health economics outcome section, which is reported separately to the clinical outcomes.

The review was conducted using several established databases (although only 3 are reported in the abstract). The attempt was for a more quantitative piece but the quality of the research has produced a narrative synthesis review focusing on three areas.

The findings include suggestions that Direct Access patients have similar characteristics as GP led services, there is a mixed type of model used, the barriers and facilitators are identifiable and again varied. Finally little difference was shown in outcomes (clinical) but limitations in design restrict such conclusions.

This paper has much to commend it; namely

• The subject is topical and is a growing area and will therefore of interest to the MSK readership.

• There is little work on this area to date.

• The work is very well executed and well written, with high levels of rigour and balance.

• All the correct instruments for systematic review have been used; it is registered on Prospero and reported according to PRISMA guidelines.

• Resolution of conflicts for each item was done appropriately.

• The work on healthcare economics is more persuasive and perhaps one of the stronger messages (acknowledged by the authorship team).

• The Discussion is reasonable and clear.

• Limitations are fairly reported.

• The Conclusion is sensible.

Despite these qualities there also some (perhaps more minor) areas of disquiet.

• The review is only a narrative review which does limit its use. (It is accepted this is not the research teams’ issue)

• There was inclusion of experimental studies only (presumably with a view to an attempt at a more quantitative piece in the first instance – which was not possible). This may have limited some rich information which would have been acceptable in a narrative review.

• Barriers and facilitators reflect largely administration issues

• One of my substantial concerns is the overall gravity and impact of the work. The first two aims are a little lacking in substance - describing characteristics of the population (just age and gender) and some chronicity does not really help practice a great deal. Similarly, describing the models available is also a little lacking in ambition. The work on outcomes has more of a research question feel and would appear to offer greater impact. However, with only 9 studies included and an inability to fully synthesise data, any useful conclusion becomes difficult (again to be entirely fair to the authors they acknowledge this).

• Some of the tables are unwieldy (I understand they are necessary to some extent for Reviews). Table 2 will be unpublishable because of the amount of whitespace and limited information provided.

• Table 3 has a great deal more information than just the type or model of self referral. Again unlikely to be publishable in the given form.

Summary:

A well-executed piece of work and well written up. It is a valuable resource of information on this subject and will help others in the area but, the impact of the work could be questioned. Systematic reviews are useful if they increase our understanding or change practice. There is some modest increase in our understanding but impact on practice is unlikely. As an information compendium and assimilation of available literature it is high quality and therefore does has value.

Reviewer #3: Your study is a succesful one and contribute the literature an important subject. I hae a few offer,

Thank you

1. You start with MSK in the abstract, please do not use abbrevation wşthout defining it before.

2.You need to define aim of the study better.,

3. Thi fügure 1 do not seem clear. Also you have an extra space between the numbers such as 13 176; 9 010. Please delete the unnecessary spaces.

4.Your reference number is few. This is a review study and I think you should use more studies.

6. PLOS authors have the option to publish the peer review history of their article (what does this mean?). If published, this will include your full peer review and any attached files.

Reviewer #1: No

Reviewer #2: No

Reviewer #3: No

---

## [Editor Report · Decision Letter 1]

15 Jun 2020

A Systematic Review and Evidence Synthesis of Non-Medical Triage, Self-Referral and Direct Access Services for Patients with Musculoskeletal Pain.

PONE-D-20-05712R1

Dear Dr. Babatunde,

We’re pleased to inform you that your manuscript has been judged scientifically suitable for publication and will be formally accepted for publication once it meets all outstanding technical requirements.

Kind regards,

Alison Rushton

Academic Editor

PLOS ONE

Additional Editor Comments (optional):

Thank you for addressing all reviewers' comments to a satisfactory level. I hope you agree that this has further improved the manuscript.
---

## [Editor Report · Acceptance letter]

17 Jun 2020

PONE-D-20-05712R1 

A Systematic Review and Evidence Synthesis of Non-Medical Triage, Self-Referral and Direct Access Services for Patients with Musculoskeletal Pain. 

Dear Dr. Babatunde:

I'm pleased to inform you that your manuscript has been deemed suitable for publication in PLOS ONE. Congratulations! Your manuscript is now with our production department. 

Kind regards, 

on behalf of

Dr. Alison Rushton 

Academic Editor

PLOS ONE